# Changed Temporal Structure of Neuromuscular Control, Rather Than Changed Intersegment Coordination, Explains Altered Stabilographic Regularity after a Moderate Perturbation of the Postural Control System

**DOI:** 10.3390/e21060614

**Published:** 2019-06-21

**Authors:** Felix Wachholz, Tove Kockum, Thomas Haid, Peter Federolf

**Affiliations:** Department of Sport Science, University of Innsbruck, 6020 Innsbruck, Austria

**Keywords:** temporal regularity, coordinative complexity, motor control, center of pressure COP, principal component analysis PCA, balance, perturbation, neuroscience

## Abstract

Sample entropy (SaEn) applied on center-of-pressure (COP) data provides a measure for the regularity of human postural control. Two mechanisms could contribute to altered COP regularity: first, an altered temporal structure (temporal regularity) of postural movements (H1); or second, altered coordination between segment movements (coordinative complexity; H2). The current study used rapid, voluntary head-shaking to perturb the postural control system, thus producing changes in COP regularity, to then assess the two hypotheses. Sixteen healthy participants (age 26.5 ± 3.5; seven females), whose postural movements were tracked via 39 reflective markers, performed trials in which they first stood quietly on a force plate for 30 s, then shook their head for 10 s, finally stood quietly for another 90 s. A principal component analysis (PCA) performed on the kinematic data extracted the main postural movement components. Temporal regularity was determined by calculating SaEn on the time series of these movement components. Coordinative complexity was determined by assessing the relative explained variance of the first five components. H1 was supported, but H2 was not. These results suggest that moderate perturbations of the postural control system produce altered temporal structures of the main postural movement components, but do not necessarily change the coordinative structure of intersegment movements.

## 1. Introduction

Human postural control is the ability to keep the human body in an upright position and stay balanced [1]. However, numerous neurological impairments, such as concussion [2], autism [3], stroke [4], cerebral infarction [5], Parkinson’s [6], a hearing deficit [7], Ehlers–Danlos syndrome [8], idiopathic scoliosis [9], cerebral palsy [10], or down syndrome [11] affect and often compromise postural control. Moreover, physiological states like fatigue, e.g., in the hip and ankle muscle [12] or the general lower extremity [13], were also shown to have an effect on postural control. The age factor has an impact on the postural control system [14], as well as alterations that affect the system by disturbing the “normal state”, like dual-tasking situations [15] or when less sensory input is received after closing the eyes [16]. Several studies report effects on postural control after rapid head movements—for example, in patients with compromised vestibular function [17], in the elderly [18], or in sports like soccer (after heading the ball) [19].

To evaluate differences in postural control, a common method is to analyze center-of-pressure (COP) data. COP data can be analyzed using different approaches, such as stabilogram diffusion [20] or using COP displacement [21]. Another method that has gained attention and validation over recent years is using sample entropy (SaEn) of COP data [22]. SaEn can be described as a measure of regularity adapted to strings of data in a time series [16,23]. Research suggests that compromised postural control systems tend to produce more regular (low-SaEn-value) movements compared to healthy postural control systems, which typically show less regularity (high SaEn-value). This was shown, for example, in patients suffering from concussion [2,24,25], Ehlers–Danlos syndrome [8], idiopathic scoliosis [9], or multiple sclerosis [26]. Moreover, evaluating COP entropy to assess postural control has, in several situations, proved to be more sensitive than traditional COP measures [2,27]. 

COP entropy is often interpreted as a measure of the regularity of the COP time series [28,29,30], and thus of the postural changes that produced the COP movement. Even during a quiet stance, several types of postural changes have been discussed, e.g., ankle, hip, [31,32] and higher-order movement strategies [33]. COP regularity could therefore be a measure for the regularity of the temporal structure of the underlying movement strategies (hypothesis 1), thus characterizing the temporal structure of the individual movement strategies. However, we could also speculate that a more complex interplay of the segment movements increases the irregularity of the COP trajectory (hypothesis 2). In the latter case, COP regularity would be a measure for the complexity of the coordinative structure of the inter-segment movements. A measure for coordinative complexity could be the dimensionality [34] of the postural movements, assessable through a principal component analysis (PCA) on kinematic movement data recorded during a balancing exercise [33,35,36,37]. In a recent study by our group [38], we investigated the two proposed hypotheses for the origins for COP regularity with a cross-sectional study design. We measured COP motion, as well as the kinematic movements of all body segments through a marker-based motion tracking system, and performed a principal component analysis on the kinematic marker data to obtain a representation of the postural movement strategies; we could thus assess the temporal regularity by calculating the entropy of the dominant movement components, as well as the dimensionality of the postural movements by calculating the cumulative variance, *rCUM_x_*, which expresses (in percent) how much the first *x* movement components contribute to the overall variance of a person’s postural variance [38] (high *rCUM_x_* indicates lower dimensionality, i.e., a less complex movement structure, while a lower *rCUM_x_* suggests that more movement strategies contributed to the overall movement, i.e., a more complex motion). We found that between subjects, COP regularity correlated with both the temporal regularity of the underlying movement components, as well as with the dimensionality of the postural movements [38]. The purpose of the current study was to investigate the two potential sources of altered COP regularity with an intervention study design. 

In summary, the current study was conducted to better understand what COP entropy, or more specifically a change in COP entropy, signifies for the actual postural movements that produced the analyzed COP trajectory. Thereto, the goal was to moderately perturb the postural control system of the volunteers through rapid, voluntary head movements, in order to produce changes in COP entropy. We could then assess the two hypotheses by determining whether changes in COP regularity were accompanied by changes in the regularity of the temporal structure in individual movement components (hypothesis 1), or by changes in the dimensionality of the overall postural movement (hypothesis 2). Two types of head movements, nodding and tilting (ear-to-sholder), were used to disturb the postural control system. Furthermore, since differences in postural control are often found between eyes open and eyes closed conditions, we also evaluated if differences can be found between performing the tests with open or closed eyes. 

## 2. Materials and Methods 

### 2.1. Ethics

Before performing the measurements, all participants were informed about the procedures and about possible risks involved. All participants gave written informed consent and were informed that they could withdraw from the experiment at any time without reason. The study was approved by the board for Ethical Questions in Science of the University of Innsbruck (52/2017) and all procedures were performed in accordance with the Declaration of Helsinki.

### 2.2. Participants

A convenience sample of 16 healthy and physically active volunteers was analyzed in the current study (see Table 1). Exclusion criteria were neurological disorders or injuries sustained within the last six months, as well as pathological joint, tendon, or muscle problems.

### 2.3. Measurement Procedures

The participants performed a 5 min warm-up before the actual measurement, which included shoulder movements and flexing neck muscles. For the actual measurements, participants were instructed to stand as steady as possible in a hip-wide stance, with their hands on the hips and their gaze focused on a sign placed 5.5 m in front them, at a height of 1.75 m. Each measurement trial lasted 130 s and consisted of three parts. For the first 30 s, the participants were instructed to stand as still as possible. Then, in response to a signal from the experimenter, the participants were to shake their head for 10 s. Completion of this part was indicated to the participants through another signal from the experimenter. The number of performed headshakes was quantified by the movement of the markers attached to the head and are displayed in Table 2. After the headshaking, participants were to continue with a quiet stance for another 90 s. The headshaking was completed at volunteers own pace and at self-chosen range of motion. Participants were instructed to, on the one hand, shake as fast as possible and, on the other hand, to do it in a way that felt safe and did not compromise standing stability or control of motion. Some volunteers reported the sensation of a light dizziness after the head shaking part in some of their trials; however, this did not occur systematically and was not further analyzed in the current study. In total, four trials were completed per participant in a randomized order. Two different methods of head shaking where conducted: in the anterior–posterior direction, comparable to a “nodding” motion; and medio-laterally, which can be described as tilting the head sideways (ear-to-shoulder). Both trials were executed once with eyes open and once with eyes closed. Hence, every participant did one trial per head shake, resulting in four trials per participant. Between the trials, participants had at least 60 s of rest in which they could walk around in the room.

### 2.4. Instrumentation

Center-of-pressure motion was recorded at 3000 Hz using a ground-embedded AMTI force plate (AMTI, Watertown, NY, USA). Kinematic data quantifying postural movements of the body were collected at 150 Hz using an eight-camera Vicon motion tracking system (Vicon Motion Systems Ltd., Oxford, UK). In total, 39 retro-reflective markers were attached to anatomical landmarks on the skin or on tight clothing, using double-sided tape. Markers on the wrists and head were attached using modified sweatbands. The marker model was based on the full-body Plug-In Gait marker model of Vicon. Force plate and kinematic data were synchronously collected using the Vicon software (Vicon Nexus, Version 2.2.3; Vicon Motion Systems Ltd., Oxford, UK). 

### 2.5. Data Processing

Kinematic and COP data were further analyzed using MatLab™ R2015b (The Mathworks Inc., Natick, MA, USA). From each trial, three 15 s phases were extracted (Figure 1). Phase 1 was taken from the period before the headshake, from second 5 to second 20. The first five seconds were omitted to evade movements related to starting the measurement. The second 15 s phase started as soon as the headshake had stopped, which was separately established for each trial, by determining when the movement of the markers placed on the head no longer exceeded quiet-standing movements. Phase 3 was selected right after the end of phase 2, since in this phase postural control had returned to normal (Figure 1), and since we wanted to avoid effects of fatigue or impatience, which were sometimes observed later in a trial.

### 2.6. Processing of Center-of-Pressure Data

The COP data was down-sampled to 150 Hz and de-trended using a moving average calculated from the surrounding 300 neighboring points of each data point. The first and last 150 data points were padded with the first and last de-trended data point, respectively. The anterior–posterior *COP_AP_*(*t*) and the medio-lateral component *COP_ML_*(*t*) of the COP were analyzed separately.

### 2.7. Processing of Kinematic Data: Calculation of Principal Movements (PMs)

The kinematic data of each phase was gap-filled [39,40], centered by subtracting the subject mean [33], normalized to the mean Euclidean distance [36,41], and weighted according to the relative weight represented by each marker [42]. The data sets of the three phases from all subjects were concatenated to create a 432,000 × 117 input matrix for the PCA (lines: 16 subjects, four trials, with three phases of 15 s recorded at 150 Hz; columns: 39 three-dimensional (3D) marker coordinates). Each row of the matrix contains the available information about the posture of a participant at a certain time point, and was interpreted as an 117-dimension posture vector [36,43,44,45,46]. After normalizing and concatenating the data from all participants and all trials, one PCA could be calculated for the whole dataset, which has the advantage that the resultant movement components can be directly compared between trials and participants. The PCA was calculated as eigenvector decomposition of the covariance matrix of the data. The PCA created a new basis (eigenvectors) in the vector space of posture vectors. PCA applied to kinematic movement data outputs eigenvectors *PC_k_* (where *k* indicates the order of the eigenvectors), which each describe one linear pattern of correlated marker motion. The PCA also outputs scores *PP_k_*(*t*) and eigenvalues *EV_k_*. The scores are a representation of the posture vectors in the new *PC_k_*-basis, and are obtained by projecting the original data on the new basis, i.e., by a basis transformation. We call the scores also “principal positions” *PP_k_*(*t*), since they represent positions in posture space [36,37,38,40]. The eigenvectors *PC_k_* and their associated principal positions *PP_k_*(*t*) together define one (the *k*-th) component of the whole postural movements. We call these movement components “principal movements” (*PM_k_*). The *EVk* are an indicator of the contribution of each *PM_k_* to the overall postural variance created by all participants [38,43,44,45,46]. To obtain an analogue, subject-specific variable, we determine from *PP_k_*(*t*) the relative variance *rVAR_k_* that each *PM_k_* contributes to the participant’s entire postural variance [33]. Further, the values of the *rVAR_k_* were cumulated to calculate the cumulative relative variance *rCUM* representing the dimensionality of the movement [47], thus providing a measure for what we described as coordinative complexity.

### 2.8. Sample Entropy Calculation

The regularity of *COP_AP_*(*t*), *COP_ML_*(*t*) and of the first 10 (*k* = 1...10) *PP_k_*(*t*) time series was determined by calculating the sample entropy (SaEn). The SaEn-algorithm was computed with the parameters’ embedding dimension *m* = 2 and tolerance *r* = 0.2·STD, where STD is the standard deviation of the time series [48]; a time delay *τ* was chosen that corresponds with physiological time-scales—specifically, *τ* was set to *τ =* 15, corresponding to 100 ms [49]. For the calculation of the regularity of the *PP_k_* time series, the same parameters were chosen. The choice of these parameters over a wide range of values did not affect the ranking of the SaEn values, nor of the conclusions that could be drawn from the subsequent statistical evaluation (see Appendix A), which agreed with earlier observations [38].

### 2.9. Statistics

Dependent variables were *SaEn(COP_AP_)*, *SaEn(COP_ML_)*, *SaEn(PP_k_)*, *rVAR_k_* and *rCUM_k_*. Shapiro–Wilk tests were used to test for normal distribution, and the Greenhouse–Geisser correction was performed if the sphericity condition was not met. A one-way, repeated-measure analysis of variance (rANOVA) was used to test for differences between phases (repeated, 3 levels). In the case of significant main effects of phases, a Sidak-corrected post-hoc test was performed to determine where differences originated. If the data was not normally distributed, separate Friedmann tests were performed—in this case, post-hoc tests were calculated using the Dunn–Bonferroni correction. All statistical testing was conducted using SPSS (IBM SPSS Statistics, Version 23); the alpha level was set to α = 0.05, and partial eta squared *η*^2^ was used to quantify the effect size.

## 3. Results

### 3.1. Center of Pressure Sample Entropy

Most differences between phases were found in the entropy of the anterior–posterior COP component (Figure 2). Sideways (tilting) head shaking produced significant phase effects with eyes open: *F*(1.432, 21.474) = 6.244, *p* = 0.013, *η*^2^ = 0.294 (post-hoc: phase 2 SaEn was significantly (*p* = 0.34) lower than phase 3 SaEn); and with eyes closed: *F*(1.478, 22.163) = 10.653, *p* = 0.001, *η*^2^ = 0.415 (post-hoc: phase 1 higher than phase 2 (*p* = 0.006), and phase 2 lower than phase 3 (*p* = 0.025). However, nodding the head did not reveal significant differences in *SaEn(COP_AP_),* neither with eyes open (*F*(1.250, 18.754) = 3.325, *p* = 0.077, *η*^2^ = 0.181) nor with eyes closed (χ^2^(2) = 4.500, *p* = 0.105). 

In the medio-lateral direction of the COP (Figure 2), significant differences were only found during nodding the head with eyes open χ^2^(2) = 10.500, *p* = 0.005 (post-hoc indicated phase 2 SaEn was different from phase 3: *p* = 0.004). Neither nodding the head with eyes closed (χ^2^(2) = 2.625, *p* = 0.269), nor sideways tilting with eyes open (*F*(1.482, 22.229) = 2.526, *p* = 0.115, *η*^2^ = 0.144), nor sideways tilting with eyes closed (χ^2^(2) = 0.875, *p* = 0.646) caused a significant phase effect in the *SaEn(COP_ML_)*.

### 3.2. Effects of Head Shaking on the Entropy of the Kinematic Movement Components

The first five PMs described 94.16% of the total movement variance in the trials. The video sequence attached as Appendix A to this paper visualizes these movement components, and Table 3 lists their eigenvalues and gives a qualitative interpretation for what aspect of the total postural movement each PM component quantified. An overview of the effect of head shaking (phase effect) on the entropy of these movement components (in the *PP_k_*-time series) is given in Table 4. In summary, in eyes-closed trials, tilting the head produced a significant change in the entropy of all of the first five movement components. In the eyes-open trials, tilting produced a significant change in the entropy of the second, third, fourth, and fifth movement components, but not in the first. Whereas the anterior-posterior head shaking produced significant changes in the four movement components (PM_1_, PM_2_, PM_3_, and PM_4_) during the eyes-open trial, but did not produce significant changes in any of the movement components during the eyes-closed trials (Table 4).

### 3.3. Effects of Head Shaking on the Coordinative Complexity of the Postural Movements

An overview of the effects of head shaking on the relative contribution of each movement component to the overall movement (*rVAR_k_*) and on the cumulated relative variances (*rCUM*) are given in Table 5 and Table 6, respectively. For each variable, only one significant result was observed in all testing situations, as well as in all of the first five movement components. This corresponds to what is expected as a coincidental result (1 in 20). The results therefore do not allow adopting the hypothesis that head-shaking had an effect on the coordinative structure or dimensionality of the postural movements.

## 4. Discussion

### 4.1. Main Results

The purpose of the current study was to determine the origin of changes in COP regularity (quantified through SaEn of the COP time series) in response to moderate perturbations of the human postural control system. We hypothesized that changes in the regularity of movement components could produce changes in COP regularity (hypothesis 1), or that changes in coordinative complexity (i.e., changes in the structure and dimensionality of the whole-body movement) would produce changes in COP regularity (hypothesis 2). Our results supported the first, but not the second hypothesis: we did find decreased COP regularity immediately after head-shaking, particularly for tilting (ear-to-shoulder) head-shaking, and specifically in the anterior–posterior component of the COP time series. For these trials, we also found decreased regularity in the time series of the first five movement components, as predicted by hypothesis 1. However, we did not find significant changes in the structure or the dimensionality of the whole-body postural movements, contrary to the prediction of hypothesis 2. Figure 3 visualizes this finding: if changes in anterior–posterior COP entropy are observed (top graph), then hypothesis 1 predicts changes, particularly in the movement components with substantial anterior–posterior contributions (PM_1_, PM_3_, PM_5_; middle three graphs), while hypothesis 2 predicts changes in the coordinative complexity (here represented through rCUM_5_; bottom graph), which were not observed.

The results of the current study disagree with an earlier study of our group, in which we investigated the same research question with a cross-sectional study design [38]. In this earlier study, we found that higher COP–SaEn across participants correlated with both higher entropy of relevant individual movement components and with higher coordinative complexity of the whole-body postural movements. Hence, while the earlier study suggested that the regularity of the control of individual movement components and the coordinative complexity are related, the current findings suggest that moderate perturbations of the sensorimotor control system primarily affect the control of movement components, but do not necessarily perturb their composition or the dimensionality in the resultant overall postural movements. In other words, the control system modifies how individual movement components are controlled (regularity of the temporal structure); it does not necessarily employ additional movement components (coordinative complexity in the sense of movement dimensionality). 

The observation that COP entropy was altered predominantly in the anterior-posterior direction is plausible, considering that the medio-lateral direction in normal bipedal stance is well-stabilized between the two legs, whereas in the anterior–posterior movement direction the neuromuscular control plays a more direct role [32,37]. The different control mechanisms in the static position might also play an important role for the observed difference. The anterior–posterior direction is mainly controlled by ankle (PM_1_) and hip (PM_3_) movements, while the medio-lateral direction is controlled mostly by shifting the weight between the legs (PM_2_), reflecting changes in stance [35].

Another observation was that head-shaking in the lateral direction (shoulder-to-ear) seemed to have a more pronounced effect on postural control than head-shaking in a nodding movement. These head-shaking directions may affect the sensory systems differently, particularly the vestibular system. An additional explanation could be that the lateral head movement produces larger strains on internal structures of the brain, particularly on the corpus callosum—a mechanism proposed in recent simulation studies on concussion [50].

### 4.2. Limitations

The participants were not screened for medication or for chronic lower back pain, which both might have an effect on postural control. 

Another important limitation of the current study is the small sample size. On the one hand, we observed statistical significance in a number of variables where it was expected (according to hypothesis 1)—a result which we believe to be reliable. On the other hand, where we did not observe significance (e.g., in the tests for hypothesis 2), the small sample size and the often relatively small effect sizes (Table 4, Table 5 and Table 6) led to a small statistical power. Hypothesis 2 is thus not conclusively disproven. Nevertheless, when comparing effect sizes, and also the graphical representations of variables referencing the two hypotheses, we feel confident to state that the hypothesis 1 mechanism was dominating the changes in COP entropy.

A third limitation is that the head-shaking experiments were relatively modest perturbations of the human postural control system. The findings of the current study can therefore not be generalized to more severe perturbations or to pathological conditions.

Limitations arising from the data analysis approach include that PCA is a linear decomposition method, and that limiting the analysis to the first five PCs represents an approximation of the true postural movements. Also, we limited the entropy calculation to one specific time lag. While multi-scale entropy may provide additional information [51,52,53], in our experience [37,39], the statistical conclusions were not sensitive to changes in time lag over a large range of physiologically sensible lags. 

## 5. Conclusions

The current study found that *COP_AP_* entropy decreased after tilting head movements, indicating a change in the control of posture. The source of these changes in COP entropy were changes in the temporal structure (entropy) of the control of individual movement components, rather than changes in the coordinative complexity (i.e., the dimensionality) of the postural movements.

## Figures and Tables

**Figure 1 entropy-21-00614-f001:**
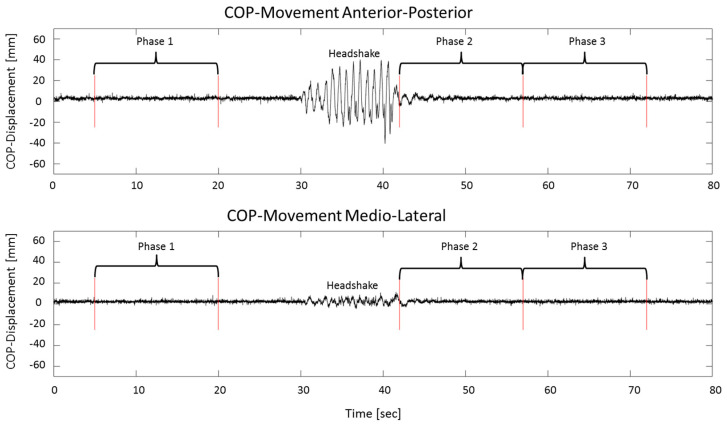
Center-of-pressure (COP) data of one sample trial of an anterior–posterior head shake with open eyes. The graph on top shows the anterior–posterior COP movement, the bottom graph shows the medio-lateral COP movement. The three analyzed phases are indicated. The last 50 s were omitted to better visualize the relevant phases of the trial.

**Figure 2 entropy-21-00614-f002:**
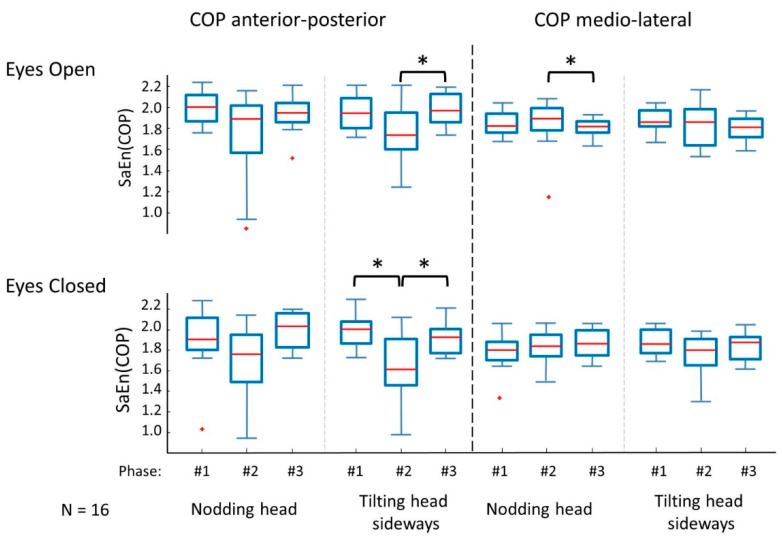
Entropy of the COP time series during 15 s of quiet standing measured before phase 1, immediately after phase 2, and later after phase 3, the 10 s head shaking exercise. Significant differences between phases are indicated with an asterisk (*).

**Figure 3 entropy-21-00614-f003:**
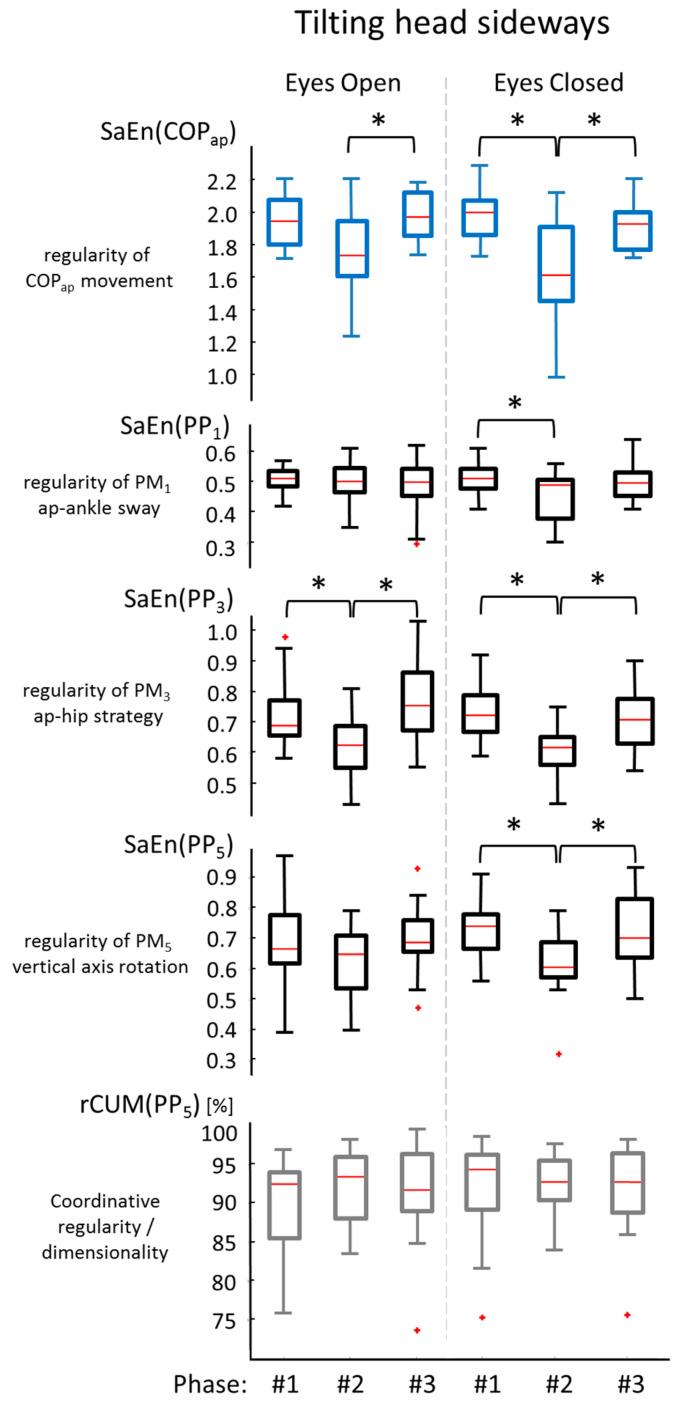
Results of the trials where participants tilted their heads sideways, with a focus on the variables representing the anterior–posterior movement of the body. The boxplots on top show the *SaEn(COP_x_)* values as seen in Figure 2. The three graphs below represent the *SaEn(PP_k_)* values of movement components substantially contributing to anterior–posterior postural movements. Significant differences are indicated (*). The bottom graph represents coordinative complexity characterized through *rCUM(PP_5_)*. No significant effects were observed in this data.

**Table 1 entropy-21-00614-t001:** Characteristics of the participants (mean ± SD).

Participants	Age	Body Weight (kg)	Body Height (cm)
All [*n* = 16]	26.5 ± 3.5	69.3 ± 14.6	176.6 ± 11.0
Female [*n* = 7]	27.1 ± 3.3	54.6 ± 5.4	165.3 ± 4.9
Male [*n* = 9]	26.0 ± 3.6	80.8 ± 7.3	185.2 ± 4.8

**Table 2 entropy-21-00614-t002:** Number of headshakes (full cycles) per second (mean ± SD).

Trial	Nodding Head	Nodding Head	Tilting Head	Tilting Head
	Eyes Open	Eyes Closed	Eyes Open	Eyes Closed
headshakes per second, averaged over all participants	2.0 ± 0.6	2.1 ± 0.6	1.8 ± 0.4	1.7 ± 0.4

**Table 3 entropy-21-00614-t003:** Qualitative description of the first five movement components (PMs) and their eigenvalues, representing each one’s contribution to the overall movement.

PM_k_	Qualitative Description of PM_k_	Eigenvalues_k_ Share to Whole Motion (%)
PM_1_	Ankle sway anterior–posterior	70.67
PM_2_	Ankle sway medio-lateral	12.19
PM_3_	Hip strategy (Hip flexion/extension) anterior-posterior	5.72
PM_4_	Lifting shoulders / Breathing	3.87
PM_5_	Upper body rotation	1.71

**Table 4 entropy-21-00614-t004:** Statistical results for the effects of head shaking (phase effect) on the entropy of the *PP_k_* time series for the four testing conditions based on repeated-measure analysis of variance (rANOVA) (black letters) or Friedmann tests (grey letters). For rANOVA results, partial eta-square (*η*^2^) values are reported as a measure of the effect size. Asterisks (*) indicate significance, and two asterisks (**) indicate highly significant head-shaking effects on the entropy of the movement component.

*PP_k_*	Trial		Overall				Post-Hoc between Phases		
			*F* | *χ*^2^	DoF	*p*-Value	*η* ^2^	1–2	2–3	1–3
***PP*_1_**	Nodding head	eyes open	10.894	2, 30	**0.001 ****	0.421	**0.001 ***	-	**0.033 ***
		eyes closed	0.213	2, 30	0.809	0.014	-	-	-
	Tilting head	eyes open	0.302	1.469, 22.028	0.675	0.020	-	-	-
		eyes closed	4.244	1.405, 21.078	**0.040 ***	0.221	**0.040 ***	-	-
***PP*_2_**	Nodding head	eyes open	5.907	2, 30	**0.007 ***	0.283	**0.034 ***	**0.030 ***	**0.024 ***
		eyes closed	2.865	2, 30	0.078	0.160	-	-	-
	Tilting head	eyes open	6.500	2	0.039 *	-	0.040 *	-	-
		eyes closed	6.175	2, 30	**0.006 ***	0.292	**0.045 ***	**0.017 ***	-
***PP*_3_**	Nodding head	eyes open	6.258	2, 30	**0.005 ***	0.294	**0.025 ***	-	-
		eyes closed	1.365	2, 30	0.271	0.083	-	-	-
	Tilting head	eyes open	9.820	2, 30	**0.001 ***	0.369	**0.022 ***	**0.007 ***	-
		eyes closed	12.310	2, 30	**0.001 ****	0.451	**0.001 ****	**0.007 ***	-
***PP*_4_**	Nodding head	eyes open	5.158	2, 30	**0.012 ***	0.256	-	-	-
		eyes closed	1.580	2, 30	0.223	0.095	-	-	-
	Tilting head	eyes open	9.745	2, 30	**0.001 ***	0.394	**0.005 ***	**0.006 ***	-
		eyes closed	13.225	2, 30	**0.001 ****	0.469	**0.003 ***	**0.003 ***	-
***PP*_5_**	Nodding head	eyes open	1.392	2, 30	0.264	0.085	-	-	-
		eyes closed	1.793	2, 30	0.184	0.107	-	-	
	Tilting head	eyes open	3.412	2, 30	**0.046 ***	0.185	-	-	-
		eyes closed	7.148	2, 30	**0.003 ***	0.323	**0.007 ***	**0.038 ***	-

**Table 5 entropy-21-00614-t005:** Statistical results for the effects of head shaking (phase effect) on the relative variance *rVAR* of *PP_k_* for the four testing conditions based on rANOVA. For rANOVA results, partial eta-square (*η*^2^) are reported as a measure of the effect size, and 1−*β* for the observed power. Asterisks (*) indicate significant head-shaking effects on the relative variance of the movement component.

*PP_k_*	Trial		Overall *rVAR_k_*				
			*F*	DoF	*p*-Value	*η* ^2^	1 − *β*
***PP*_1_**	Nodding head	eyes open	1.320	2, 30	0.282	0.081	0.263
		eyes closed	1.719	2, 30	0.196	0.103	0.332
	Tilting head	eyes open	0.625	2, 30	0.542	0.040	0.145
		eyes closed	0.274	2, 30	0.763	0.018	0.089
***PP*_2_**	Nodding head	eyes open	1.228	1.480, 22.201	0.300	0.076	0.213
		eyes closed	6.387	2, 30	**0.005 ***	0.299	0.870
	Tilting head	eyes open	2.023	2, 30	0.150	0.119	0.384
		eyes closed	0.473	2, 30	0.628	0.031	0.120
***PP*_3_**	Nodding head	eyes open	2.691	2, 30	0.084	0.152	0.492
		eyes closed	0.977	2, 30	0.388	0.061	0.204
	Tilting head	eyes open	1.418	2, 30	0.258	0.086	0.280
		eyes closed	0.093	2, 30	0.912	0.006	0.063
***PP*_4_**	Nodding head	eyes open	0.601	2, 30	0.555	0.039	0.141
		eyes closed	0.294	2, 30	0.748	0.019	0.092
	Tilting head	eyes open	0.175	2, 30	0.840	0.012	0.075
		eyes closed	0.737	2, 30	0.487	0.047	0.163
***PP*_5_**	Nodding head	eyes open	0.726	1.415, 21.218	0.450	0.046	0.141
		eyes closed	0.574	2, 30	0.569	0.037	0.136
	Tilting head	eyes open	0.583	2, 30	0.564	0.037	0.138
		eyes closed	0.476	2, 30	0.626	0.031	0.121

**Table 6 entropy-21-00614-t006:** Statistical results for the effects of head shaking (phase effect) on the relative cumulative variance *rCUM* of *PP_k_* for the four testing conditions based on rANOVA (black letters) or Friedmann tests (grey letters). For rANOVA results partial eta-square (*η*^2^) are reported as a as measure of the effect size, and 1−*β* for the observed power. Asterisks (*) indicate significant head-shaking effects on the relative variance of the movement component.

*PP_k_*	Trial		Overall *rCUM*				
			*F* | *χ*^2^	DoF	*p*-Value	*η* ^2^	1 − *β*
***PP*_1_**	**Nodding head**	eyes open	1.320	2, 30	0.282	0.081	0.263
		eyes closed	1.719	2, 30	0.196	0.103	0.332
	Tilting head	eyes open	0.625	2, 30	0.542	0.040	0.145
		eyes closed	0.274	2, 30	0.763	0.018	0.089
***PP*_2_**	Nodding head	eyes open	3.917	2, 30	**0.031 ***	0.207	0.661
		eyes closed	0.032	2, 30	0.968	0.002	0.054
	Tilting head	eyes open	1.004	2, 30	0.378	0.063	0.208
		eyes closed	0.206	2, 30	0.815	0.014	0.079
***PP*_3_**	Nodding head	eyes open	2.816	2, 30	0.076	0.158	0.551
		eyes closed	0.524	2, 30	0.597	0.034	0.128
	Tilting head	eyes open	0.765	2, 30	0.474	0.049	0.168
		eyes closed	0.178	2, 30	0.838	0.012	0.075
***PP*_4_**	Nodding head	eyes open	2.822	2, 30	0.075	0.158	0.512
		eyes closed	0.500	2	0.779	-	-
	Tilting head	eyes open	0.966	2, 30	0.392	0.060	0.202
		eyes closed	0.141	2, 30	0.869	0.009	0.070
***PP*_5_**	Nodding head	eyes open	2.823	2, 30	0.075	0.158	0.513
		eyes closed	1.625	2	0.444	-	-
	Tilting head	eyes open	0.876	2, 30	0.427	0.055	0.187
		eyes closed	2.625	2	0.269	-	-

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
