# Peer review of "Changed Temporal Structure of Neuromuscular Control, Rather Than Changed Intersegment Coordination, Explains Altered Stabilographic Regularity after a Moderate Perturbation of the Postural Control System"

_entropy, 2019, doi:10.3390/e21060614_

Round 1

Reviewer 1 Report

I was unable to open the supplementary zip file. 

General comments

1.  Sample entropy does not reflect complexity. This is an error in the literature that should be rectified. An increase in entropy moves closer to white noise or randomness. Randomness is not complex, it is complicated. There is a nice paper by Vivien Marmelat and Deligneres that provides clarification on the difference between complexity and complication. For something to be complex, it need to be deterministic. Sample entropy cannot tell you whether an increase in value is from a deterministic origin or not. It is tells you that the probability that like information will appear again is low. The authors do acknowledge this and call it regularity in lines 45-48 but then call it complexity.

See Delignieres and Marmelat (2012, Critical Reviews in Biomedical Engineering) and Raffalt et al. (2018, Computers in Biology and Medicine) 

To really examine temporal complexity, the authors should consider using multiscale entropy. This will give you the change in regularity over multiple time scales, and thus could be interpreted as temporal complexity.

2.  The purpose is stated; however, it is not clear as to why the study needs to be done and why head movements were selected. Additional justification to support the purpose is needed. Currently it is stated without justification.

Further, there is no justification for the eyes open, eyes closed conditions in the introduction. It was not mentioned in the purpose either.

Additionally, after reading the discussion, it is really not clear what the purpose was. The authors state, "The purpose of the current study was to investigate the two potential sources of COP complexity with an intervention study design. ... We could then assess the two hypotheses by determining whether changes in COP complexity were accompanied by changes in the complexity of the temporal structure of individual movement components (hypothesis 1) or by changes in the coordinative complexity (hypothesis 2)." I am unsure as to how hypothesis 1 was accomplished when changes in COP complexity were measured by COP complexity. A variable cannot be both an independent and a dependent variable.

3. The discussion does not appear to discuss control of AP and ML balance in significant detail (one sentence is all that is found). In a static position, AP is controlled by the ankle hip moment couple and ML is mainly controlled by stance. This is a potential explanation of the findings that has not been explored fully.

"Another observation was that head-shaking in lateral direction (shoulder-to-ear) seemed to have a more pronounced effect on postural control than head-shaking in a nodding movement." Why is the moment that this would create and then the trunk control to compensate is not discussed.

4. Were subject head movements recorded and quantified? How do you know they were "moderate" perturbations? This is a potential confounding variable and should be quantified as well. 

Further, how do you know that these were "complex movement components"? If the complexity of the head movements were not quantified, what basis is that statement made from?

Specific comments

1. Was there a sample size justification for this study? In addition were the subjects screened for medications that may make them dizzy? What about chronic low back pain? Or was this considered a pathological joint, muscle, tendon problem?

2. Why were data detrended if they are to be used for entropy? Many movement variability scientists believe that all of the data are needed to truly capture the structure of the movement. Possibly a downsample is needed as 150Hz may introduce redundant data points for comparison in sample entropy, leading the values to be smaller than they should be. Was a power spectral analysis completed to ensure this is the correct sampling rate? See Stergiou's Innovative Analysis text, chapter 9. Also see a recent paper by Yentes et al (in Entropy and the one by Raffalt in Med Biol Eng Comput) that demonstrates how sampling rates have an effect on sample entropy values. Potentially this issue was resolved by using tau but should be verified.

3. Using an r value set by previous authors is not an appropriate approach to your data. Please confirm this is the appropriate r value for these data by checking the relative consistency. The same for tau. Are the parameters confirmed to be correct for your dataset? You reference a paper that these parameters do not change the findings but confirmation should be transparent. This confirmation should be added to supplementary data.

4. Additional information is needed as to why sample entropy was used on the principal positions and how this was done. Same parameters as the COP?

5. Please generate a figure similar to Figure 2 for ML direction. Actually Figure 2 could become a four panel figure with all data included.

6. Why is rVAR introduced as a dependent variable in section 3.3 when it was not listed in the statistical analysis section?

Author Response

Dear Editor, dear Reviewer,

Thank you for the opportunity to revise our manuscript and for the constructive comments. We believe that we could address all of these comments and have revised our manuscript accordingly. With this letter we would like to submit it for a reevaluation. 

Please find attached a point-by-point response to each reviewer comment attached.

Thank you for your time and your valuable advice.

On behalf of all co-authors,

sincerely,

Felix Wachholz

Reviewer 2 Report

A review of Entropy-520156

Changed temporal structure of neuromuscular control rather than changed intersegment coordination explains altered stabilographic complexity after a moderate perturbation of the postural control system

General Comments:

The use of sample entropy in understanding the human postural control system is well defined. The current study offers additional data analyzing sample entropy applied to center of pressure postural sway responses when exposed to postural perturbations involving rapid head shakes. This paper can offer its findings to be applied to multiple populations in addition to the clinical populations mentioned in the introduction, such as athletic, geriatric, occupational etc.

The paper is well written, and the supportive document of the marker-based trial provides a nice touch to the paper review process, in comparison to other papers with hardly any pictures (for both ease of understanding and replicating research).

I have highlighted a few questions, suggestions and comments below under the specific comments section. The introduction and discussion are well explained and justified. All my specific comments are limited to the results and methodology.

Specific Comments:

1.       60 seconds is usually the time for static postural stability tasks. While a 30 second and a 90 second follow-up trial is acceptable, why was a 10-second head shake time chosen? Any previous supporting literature that can be cited here?

2.       How many trials per head shake (nodding and sideways) were performed by each participant?

3.       Any participants wearing glasses or corrective lens, which might make the rapid head shake response even more cumbersome.

4.       Any familiarization? Especially in head shake? How is a non-familiarized rapid head shake be assumed similar among all 16 participants?

5.       Why are graphs (Figure 1) only showing 80 second trials?

6.        Can you please add a similar figure to figure1 for side-to-side head shaking with both AP and ML COP traces?

7.       Was figure 1 in eyes open or closed. Please add information.

Author Response

(The authors gave the same response as above.)

Round 2

Reviewer 1 Report

Thank you for your detailed responses. Please make the following minor changes:

Add to the limitations the lack of screening for medications and chronic low back pain as these can be common in healthy, young, active adults.

Please use Figure 2 as presented in the response to reviewers. This provides context to both the AP and ML directions.

Please add to either the methods or the discussion an acknowledgment that although the findings are significant, the effect sizes are quite small from the majority of significant findings in tables 4-6. 

Author Response

Dear Editor, dear Reviewers,

Thank you once more for the constructive comments and precise feedback. We implemented the minor suggestions and believe that we could address the comments. With this letter we would like to re-submit it for a reevaluation. Please find a point-by-point response to each reviewer suggestions attached.

Thank you again for your time and your very constructive feedback.

On behalf of all co-authors, 

sincerely,

Felix Wachholz

Reviewer 2 Report

The authors have addressed my comments. 

Author Response

Dear Editor, dear Reviewers,

Thank you once more for the constructive comments and precise feedback. 

Thank you again for your time and your very valuable feedback.

On behalf of all co-authors,  

sincerely,

Felix Wachholz